# Genetic Screening for Hereditary Transthyretin Amyloidosis in the Population of Cammarata and San Giovanni Gemini Through Red Flags and Registry Archives

**DOI:** 10.3390/brainsci15040365

**Published:** 2025-03-31

**Authors:** Vincenzo Di Stefano, Christian Messina, Antonia Pignolo, Fiore Pecoraro, Ivana Cutrò, Paolo Alonge, Nicasio Rini, Umberto Quartetti, Vito Lo Bue, Eugenia Borgione, Filippo Brighina

**Affiliations:** 1Department of Biomedicine, Neuroscience and Advanced Diagnostics (BiND), University of Palermo, 90129 Palermo, Italy; chry.messina@gmail.com (C.M.); antonia.pignolo@unipa.it (A.P.); pecorarofiore@gmail.com (F.P.); ivanacutro1993@gmail.com (I.C.); alongep95@gmail.com (P.A.); nicasio.rini@gmail.com (N.R.); umberto.quartetti@unipa.it (U.Q.); vito.lobue@policlinico.pa.it (V.L.B.); filippobrighina@gmail.com (F.B.); 2Oasi Research Institute-IRCCS, Via Conte Ruggero 73, 94018 Troina, Italy; eborgione@oasi.en.it

**Keywords:** hereditary transthyretin amyloidosis, ATTRv-PN, TTR amyloid, genetics, screening, neuromuscular, neuropathy, neurophysiology

## Abstract

**Introduction**: Hereditary transthyretin amyloidosis (ATTRv) is a severe, multisystemic, autosomal dominant disease with variable penetrance caused by mutations in the *TTR* gene generating protein misfolding and accumulation of amyloid fibrils. The diagnosis is usually challenging because ATTRv may initially manifest with nonspecific multisystemic symptoms. Conversely, an early diagnosis is needed to start timely appropriate therapy. Hence, screening models have been proposed to improve ATTRv diagnosis. In this study, we propose a genetic screening model based on predefined “red flags” followed by “cascading screening” on first-degree relatives of patients who tested positive. **Materials and methods**: After obtaining written informed consent, genetic testing on salivary swabs was performed in individuals who met at least two major red flags for ATTRv (age > 65 years old, progressive sensory or sensorimotor neuropathy not responsive to steroids or immunomodulant therapies, recent and unexplained weight loss associated with gastrointestinal signs and symptoms, diagnosis of cardiac amyloidosis, bilateral or relapsing carpal tunnel syndrome, unexplained autonomic dysfunction) or one major flag and two minor flags (family history of neuropathy, ambulation disorders or cardiopathy, sudden cardiac death, a bedridden, wheelchaired patient without specific diagnosis excluding upper motor neuron diseases, infections, juvenile cardiac disease, ocular disorders, lumbar spine stenosis, biceps tendon rupture). **Results**: In the first screening phase, 29 suspected cases (individuals meeting at least two major red flags or one major red flag and two minor red flags) underwent genetic testing. One patient (3.5%) was diagnosed with hereditary transthyretin amyloidosis with polyneuropathy (ATTRv-PN), carrying the Phe64Leu mutation. Then, cascade screening allowed for early recognition of two additional individuals (two pre-symptomatic carriers) among two first-degree relatives (100%). The identified patient was a 72-year-old man who had a family history of both cardiopathy, neuropathy, and a diagnosis of juvenile cardiac disease and progressive sensorimotor neuropathy unresponsive to steroids or immunomodulant therapies. **Conclusions**: ATTRv is a progressive and often fatal disease that should be promptly diagnosed and treated to stop progression and reduce mortality. Systematic screening for ATTRv yielded increased recognition of the disease in our neurological clinic. A focused approach for the screening of ATTRv-PN could lead to an earlier diagnosis and identification of asymptomatic carriers, enabling timely intervention through close clinical monitoring and early treatment initiation at symptom onset.

## 1. Introduction

Hereditary transthyretin amyloidosis (ATTRv) is an adult-onset, rare, and multisystemic disease, affecting mainly the sensorimotor and autonomic functions, as well as the heart, along with other organs, such as the gastrointestinal tract, eyes, and kidneys [1]. It is caused by the accumulation of abnormal amyloid fibrils originating from mutations in the *TTR* gene in an autosomal dominant inheritance pattern with incomplete penetrance. Over 150 different mutations in *TTR* have been previously described in the literature and online databases, such as the Mutations in Hereditary Amyloidosis Database (http://www.amyloidosismutations.com/, accessed on 13 January 2025), and clinical manifestations often correlate with the specific mutation [2]. Certain mutations predominantly exhibit a neurological phenotype, while others primarily manifest with a cardiac onset [2]. Therefore, clinical phenotypes are heterogeneous, often leading to a very difficult diagnosis. Consequently, misdiagnosis of ATTRv results in high costs for the healthcare system in terms of mortality and inappropriate treatments [3,4].

ATTRv amyloidosis was initially considered a disease confined to endemic foci in Portugal, Japan, and Sweden. However, the presence of late-onset cases in non-endemic areas has now been recognized, indicating that ATTRv amyloidosis is more common than previously thought [5].

The prognosis for ATTRv is going to change with the availability of new and emerging treatments. Nowadays, ATTRv appears to be more prevalent than previously thought, and there is growing emphasis on the need for early diagnosis because several treatments are most effective if started in earlier stages of the disorder, changing the disease trajectory [2]. Diagnosis of ATTRv is challenging, as it often initially presents with nonspecific symptoms, primarily affecting the nervous and cardiovascular systems. Today, physicians can rely on advanced diagnostic techniques, including echocardiographic strain imaging, magnetic resonance imaging (MRI), and nuclear scintigraphy, as well as the increased availability of genetic testing, to facilitate early and accurate diagnosis [2]. A screening model based on a “suspicion index” has been recently proposed to improve ATTRv diagnosis by Adams et al. [3]. In 2015, Mazzeo et al. estimated a prevalence of ATTRv in Sicily of 8.8/1,000,000, reporting the absence of the common Val30Met mutation and the presence of three TTR variants (Glu89Gln, Phe64Leu, Thr49Ala) [6]. In that paper, the Glu89Gln variant was quite common in eastern Sicily and Phe64Leu was common in northern Sicily, while Thr49Ala was only reported in the south, more precisely from Agrigento. However, a recent study has identified additional *TTR* mutations in western Sicily, including His90Asn, Val122Ile, Ser77Phe, and Val20Ala [6]. Of interest, the presence of Phe64Leu was confirmed in 65% of positive screening cases, and His90Asn, Val122Ile, Ser77Phe, and Val20Ala variants were also reported. Conversely, Thr49Ala and Glu89Gln were not reported. Hence, Phe64Leu seems to be the most common variant circulating in Sicily, with 28 cases reported by Mazzeo in 2015 and a further 34 cases reported by Di Stefano in 2023. Considering Italian data, Phe64Leu was also the most frequent TTR variant (54 out of 181 patients, 29.8%) reported in a large population treated with Patisiran from Italy [7]. Recently, a novel ATTRv variant, the Glu61Ala variant, has been identified in Sicily in a family from Palazzolo Acreide. These screening programs are crucial not only for detecting patients who remain misdiagnosed and, therefore, lack an appropriate diagnosis and treatment, but also for identifying new variants that have not yet been characterized [8]. Consequentially, ATTRv might have an increased prevalence in the south of Italy [6,9,10]. Cammarata is a Sicilian municipality in the province of Agrigento that has 5839 inhabitants; it is surrounded by woods and on the slopes of Cammarata mountain. San Giovanni Gemini is an enclave of Cammarata with 7463 inhabitants. We hypothesize that orogeographic, cultural, and historical characteristics of the populations of the municipalities of Cammarata and San Giovanni Gemini might have determined a functional segregation, potentially facilitating the circulation of specific *TTR* variants within this limited area. This phenomenon could be further favored by the late onset of ATTRv symptoms (generally over 65 years) and the incomplete penetrance of the disease. From this perspective, we hypothesize that specific variants in the *TTR* gene might circulate in functionally segregated populations, thus causing an apparently increased incidence of the disease. In this paper, we propose a model for genetic screening in a limited area to identify each person presenting ATTRv red flags to perform diagnostic genetic testing for diagnostic confirmation.

## 2. Materials and Methods

This is a prospective and observational study to investigate the distribution of the prevalence of ATTRv in the municipalities of Cammarata and San Giovanni Gemini. Systematic genetic screening for ATTRv was proposed in patients presenting with a suspicion index. We identified the territories of Cammarata and San Giovanni Gemini in the province of Agrigento in Sicily to perform genetic screening for ATTRv. According to available ISTAT data, the population consisted of 7463 inhabitants in San Giovanni Gemini and 5839 in Cammarata, for a total of 13,302 individuals. First, on the 11 November 2023, we held a convention to raise awareness among general physicians about ATTRv amyloidosis. Then, on the 16 March 2024, we performed systematic screening by offering diagnostic genetic testing for each individual who met at least two of the following major red flags or one major red flag and two minor red flags, which are similar to those previously reported [6,9,11]:⮚Major red flags:
○Age > 65 years;○Progressive sensory or sensorimotor neuropathy not responsive to steroids or immunomodulant therapies;○Recent and unexplained weight loss associated with gastrointestinal signs and symptoms (diarrhea, constipation, etc.) not related to changes in dietary habits;○Diagnosis of cardiac amyloidosis;○Bilateral carpal tunnel syndrome or relapsing tunnel carpal syndrome;○Unexplained autonomic dysfunction (orthostatic hypotension, recurrent episodes of loss of consciousness, erectile dysfunction, sweating impairment, etc.).⮚Minor red flags:○Family history of neuropathy or ambulation disorders;○Family history of cardiopathy or sudden cardiac death;○Bedridden or wheelchaired patient without specific diagnosis, excluding upper motor neuron diseases or infections;○Juvenile cardiac disease;○Lumbar spine stenosis;○Biceps tendon rupture;○Juvenile ocular disorders, such as glaucoma or vitreous opacity.

Exclusion criteria were the following:⮚Age < 18 years;⮚Previous ATTRv diagnosis;⮚Neuropathy responsive to steroids or immunomodulant therapies;⮚Weight loss due to voluntary or known factors;⮚Ambulation impairment due to upper motor neuron disease, infections, or other pre-existing conditions compatible with the clinical status.

As a major red flag, age > 65 years was chosen because in the previous study, mean age in screening-positive patients was >65 [6]. Indeed, onset of the disease typically occurs after 65 years for non-endemic countries. Cardiac amyloidosis was diagnosed according to standard criteria and guidelines in the case of suggestive features on scintigraphy, cardiac MRI, echo, or biopsy [12]. Before genetic testing, written informed consent was obtained from each patient. DNA was extracted from a swab sample collected from each individual according to the above-mentioned criteria. To identify mutations in the TTR gene (RefSeq NM_000371.4), PCR primers (available upon request) were designed using the software Vector NTI Advance 10.3.0 (Informax Frederick, Bethesda, MD, USA) to amplify exons 2, 3, and 4 with their flanking intronic regions. PCR reactions were carried out in 50 μL reaction volumes containing 200 ng of genomic DNA, 1X PCR reaction buffer, 0.2 mM of each dNTP, 1 μM of each primer, and 1.25 units of Taq DNA polymerase (Roche, Mannheim, Germany). The following PCR cycling conditions were used for all exons: an initial denaturation step at 94 °C for 6 min, followed by 35 cycles of 30 s at 94 °C, 30 s at 55 °C, 1 min at 72 °C, and a final extension step at 72 °C for 10 min.

PCR products were sequenced using the BigDye Terminator v1.1 Cycle Sequencing Kit (Thermo Fisher Scientific, Vilnius, Lithuania) on a SeqStudio Flex Genetic Analyzer (Applied Biosystems, Foster City, CA, USA). Patient sequence data were aligned for comparison with corresponding wild-type sequences.

This method allowed for the detection of small point mutations, deletions, or insertions (<20 bp) in the above-mentioned gene. After about four weeks, each individual received their test result, which could be either negative or positive. For each positive case, we investigated the patient’s pedigree to screen the whole family, if possible. In the case of a positive genetic test result, analysis of records from the registry office of the municipality was used to find connections between families. The mean and the standard deviation were calculated for age, while percentages were calculated for other variables.

## 3. Results

Twenty-nine individuals were identified by general practitioners who met at least two major red flags or one major red flag and two minor red flags. These patients were invited to undergo genetic testing, and none refused the swab test or denied informed consent. Among the tested patients, 28 subjects were reported to be negative (96.5%), whereas 1 patient (3.5% of those screened) presented a positive genetic test, carrying the c.250T>C (p.Phe84Leu) heterozygous variant in exon 3 of *TTR* gene, commonly referred to as Phe64Leu using legacy nomenclature. This variant is classified as “pathogenic” according to ACMG/AMP 2015 standards and guidelines and reported in the ClinVar database (Variation ID: 13453) with pathogenic clinical significance. Features of positive ATTRv patients and negative cohorts are reported in Table 1.

The patient was a 72-year-old man with a family history of both cardiopathy and neuropathy who had been diagnosed with juvenile cardiac disease and progressive sensorimotor neuropathy unresponsive to steroids or immunomodulant therapies. Among patients who tested negative, only 10 (35.7%) patients were males over 65 years old. Furthermore, 23 patients (82.1%) had neuropathy of uncertain etiology, 19 (67.9%) had carpal tunnel syndrome, 14 (50%) reported a family history of cardiopathy, 8 (28.6%) had unexplained autonomic dysfunction, 8 (28.6%) had signs of cardiac amyloidosis, 6 (21.4%) reported a family history of neuropathy, 4 (14.3%) had a diagnosis of juvenile ocular disorders, 1 individual (3.6%) experienced unexplained weight loss, and 1 (3.6%) had a diagnosis of lumbar spine stenosis. None had a diagnosis of cardiac amyloidosis or previous biceps tendon rupture. Moreover, none were bedridden or wheelchaired.

### 3.1. Family Tree and Genetic Counselling

After the diagnosis of ATTRv amyloidosis, genetic counselling was performed for the proband’s offspring. Hence, due to the high probability (50%) of being a carrier of the TTR variant, both son and daughter accepted genetic testing, resulting in them being positive for the Phe64Leu TTR variant. However, they were asymptomatic for hATTR. The proband’s parents were deceased, making it impossible to evaluate them.

### 3.2. Investigation at the Registry Office of the Municipality of Cammarata

Because a patient with ATTRv carrying the same Phe64Leu mutation as the proband was already being followed at our neuromuscular disease center, and given that this patient originated from Cammarata, we further investigated more carefully the family history to identify a common ancestor to be able to trace the origins of the mutation and a possible founder effect in the municipality of Cammarata. This research allowed us to demonstrate that both patients had a common ancestor born in the nineteenth century, as shown by an extended family tree (Figure 1).

Figure 1 shows the connection between the proband who had a positive result from the ATTRv screening genetic test and the patient carrying the same mutation already followed at our institution at the University of Palermo. This discovery demonstrated a common origin from an ancestor from the nineteenth century. The common ancestor was born in 1899; she was the proband’s aunt on the paternal side and also the grandmother of the other patient native of Cammarata. Therefore, 1899 represents the minimum dating of the existence of Phe64Leu in the city of Cammarata.

## 4. Discussion

This study systematically evaluated the impact of clinical “red flags” described in diagnostic algorithms and guidelines on predicting ATTRv diagnosis in a real-life setting. Genetic screening was offered to all patients presenting with sensory or sensorimotor polyneuropathy associated with one or more clinical feature suggestive of multisystemic ATTRv. The main result that should be underlined is that ATTRv is not as rare as previously thought, and even small municipalities, such as Cammarata, may present cases of ATTRv. As expected, the Phe64Leu variant, which is the most commonly reported variant in Sicily, was the mutation identified in this study. Based on the family tree, the proband’s grandfather was probably affected, thus suggesting that the Phe64Leu variant might have circulated in Cammarata since the nineteenth century. Moreover, the onset symptoms and signs of ATTRv can be unclear and initially attributed to other more common disorders, delaying diagnosis and treatment and leading to progressive clinical impairment [13]. Over the years, to obtain an early diagnosis and tempestive treatment, different screening models have been proposed and executed. For example, to diagnose cancer predisposition syndromes or prevent neoplastic complications, universal newborn genetic screening and general population-based screening, respectively, have been implemented [14]. Moreover, certain studies have proposed screening of the whole population based on biomarkers, with significative cost reductions and sensitivity increases [15,16]. For early diagnosis of several genetic diseases, like autosomal dominant polycystic kidney disease (ADPKD), ongoing surveillance through clinical status, blood tests, and imaging techniques has been proposed [17]. According to the screening model based on a “suspicion index” proposed by Adams et al. [3] in 2021, genetic testing should be performed in the following cases: in endemic areas, in the presence of progressive sensory length-dependent polyneuropathy or autonomic dysfunction plus at least one red flag (family history of ATTRv, unexplained weight loss ≥ 5 kilos, heart rhythm disorders, vitreous opacities, and renal abnormalities); and in non-endemic areas in case of idiopathic rapidly progressive sensory–motor axonal neuropathy or atypical Chronic Inflammatory Demyelinating Polyneuropathy (CIDP) plus at least one red flag (family history, autonomic dysfunction, early gait disorders, gastrointestinal disturbances, carpal tunnel syndrome or previous surgery for bilateral carpal tunnel, concurrent cardiac abnormalities, unexplained weight loss, vitreous opacities, and renal abnormalities) [3,18]. With our work, we propose population screening based on red flags at a specific moment. This method involves identifying individuals with at least two major red flags or one major red flag plus two minor red flags from the entire population. Once a positive case is identified, genetic testing is extended to the entire family. Diagnostic follow-up in relatives should be based on the Predicted Age of Disease Onset (PADO) score [19]. Screening based on red flags is important to improve the sensibility of the screening and to lead the patient toward the correct clinical and diagnostic follow-up while lowering costs. The main limitations of this study are the small number of screened individuals and the restricted limits of the geographical area, which could determine a selection bias, as well as the nonspecific clinical clues for ATTRv. Larger-scale screenings are needed to further validate this approach. In our study, it was essential to consult the documents present in the municipality registry archives. We were able to trace precisely the relationships between the family members and identify the relatives who represented the union in the dynasty of the two families. Through this work, we overcame the limits of traditional family anamnesis considering the considerable difficulty in recalling information from relatives beyond the first degree of kinship and in such a way as to limit errors. Ultimately, in our opinion, the consultation of registry archives should always be considered for the early diagnosis of autosomal dominant diseases.

As already mentioned, by using the registry archives, it was possible to determine the minimum dating of the existence of Phe64Leu in the city of Cammarata. The Phe64Leu variant, also known as Phe84Leu, is very rare. It has been reported in the literature with a minor allelic frequency (MAF) of 0.0000041 and a prevalence of 1:123,106 from the gnomAD database, and it is characterized by a high male-to-female ratio [20]. In the THAOS study, Phe64Leu was identified in Brazil, Argentina, the United States, and Italy [21]. In Italy, the Phe64Leu variant has been found with a higher frequency in the south, particularly in Sicily. It is commonly responsible for a late-onset mixed phenotype, with cardiological and neurological involvement, and it is particularly characterized by neurological features. Among the neurological symptoms, carpal tunnel syndrome (CTS) is frequently reported as an early sign of disease onset [18].

## 5. Conclusions

ATTRv is a progressive and often fatal disease that should be promptly diagnosed and treated to stop progression and reduce mortality. Systematic screening for ATTRv yielded increased recognition of the disease in our neurological clinic. A focused approach for the screening of ATTRv-PN could lead to an earlier diagnosis and identification of asymptomatic carriers, enabling timely intervention through close clinical monitoring and early treatment initiation at symptom onset.

## Figures and Tables

**Figure 1 brainsci-15-00365-f001:**
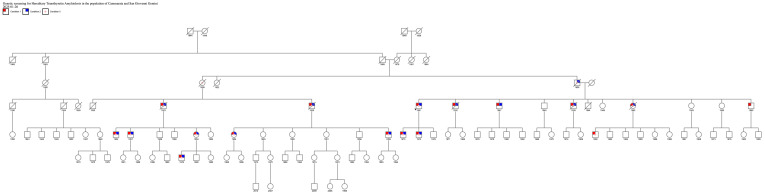
Family tree of two related patients from Cammarata. Legend: condition 1 cardiomyopathy; condition 2 neurological involvement; condition 3 ATTRv positivity.

**Table 1 brainsci-15-00365-t001:** Anamnestic and clinical findings in our reported positive testing patient (left) versus in our negative cohort (right).

Clinical Features	Positive Patient	Negative Cohort
Age (years)	72	62.3 ± 8.3
Gender	Male	13 males (44.8%)
ATTRv genotype mutation	Phe64Leu	-
Age > 65 years old	Yes	10 (35.7%)
Progressive sensory or sensorimotor neuropathy not responsive to steroids or immunomodulant therapies	Yes	23 (82.1%)
Recent and unexplained weight loss associated with gastrointestinal signs and symptoms not related to changes in dietary habits	No	1 (3.6%)
Diagnosis of cardiac amyloidosis	No	0 (0%)
Bilateral carpal tunnel syndrom or relapsing tunnel carpal syndrome	No	19 (67.9%)
Unexplained autonomic dysfunction	No	8 (28.6%)
Family history of neuropathy or ambulation disorders	Yes	6 (21.4%)
Family history of cardiopathy or sudden cardiac death	Yes	14 (50%)
Bedridden or wheelchaired patient without specific diagnosis, excluding upper motor neuron diseases or infections	No	0 (0%)
Juvenile cardiac disease	Yes	8 (28.6%)
Lumbar spine stenosis	No	1 (3.6%)
Biceps tendon rupture	No	0 (0%)
Juvenile ocular disorders	No	4 (14.3%)

## Data Availability

The data presented in this study are available upon request from the corresponding author (the data are not publicly available due to privacy restrictions).

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
