# Peer review of "Genetic Screening for Hereditary Transthyretin Amyloidosis in the Population of Cammarata and San Giovanni Gemini Through Red Flags and Registry Archives"

_brainsci, 2025, doi:10.3390/brainsci15040365_

Round 1

Reviewer 1 Report

Comments and Suggestions for Authors

With the introduction of TTR silencers/stabilizers in the last decade, it is of great importance to identify the under-recognized population of TTRv patients. Few studies have assessed screening the general population for mutations based on risk factors. While many neurologists have their own barometers of when to do genetic testing, its’ helpful to think about this screening in a more systematic way. It is even more important since serum biomarkers are lacking. While one positive genetic test out of 29 may seem relatively small, when you think of all the family members that may then be diagnosed after initial diagnosis, the impact much more. The study design was creative, particularly with educating general physicians with a convention—more of this is needed in the U.S. It would be helpful to develop a screening test that can be widely distributed to larger population of people and in other endemic areas!

A few questions:

  • While mentioned in the intro, the common mutations in seen in sicily, it would helpful to know the prevalence of each mutation that currently exists.
  • It would helpful to know why the age >65 was chosen.
  • Spontaneous tenon ruptures are quite rare in TTRv, and more common in wtTTR, why was this chosen?
  • How was cardiac amyloidosis made in the ‘major red flag’ section- PYP? Cardiac MRI? Endomyocardial biopsy?
  • Did patients that met inclusion criteria have other labs drawn including prealbumin and neurofilament light chain or were they sent directly for genetic testing?
  • Any comment about this in the algorithm to working these patients up, particularly when genetic testing may be as readily available?

Comments on the Quality of English Language

This can be improved

Author Response

Dear Editors and Reviewers,

thanks for your valuable and constructive comments. We would like you will evaluate our revised version of the manuscript for possible publication in your journal.

Reviewer #1: With the introduction of TTR silencers/stabilizers in the last decade, it is of great importance to identify the under-recognized population of TTRv patients. Few studies have assessed screening the general population for mutations based on risk factors. While many neurologists have their own barometers of when to do genetic testing, its’ helpful to think about this screening in a more systematic way. It is even more important since serum biomarkers are lacking. While one positive genetic test out of 29 may seem relatively small, when you think of all the family members that may then be diagnosed after initial diagnosis, the impact much more. The study design was creative, particularly with educating general physicians with a convention—more of this is needed in the U.S. It would be helpful to develop a screening test that can be widely distributed to larger population of people and in other endemic areas!

A: Thank you for your constructive and valuable comments. We believe that screening procedures are very important to detect significant genetic variants as well as to ensure early treatments.

While mentioned in the intro, the common mutations in seen in sicily, it would be helpful to know the prevalence of each mutation that currently exists.

A: Thank you for your interesting comment. We have cited in the introduction the TTR variant reported in Sicily. In the paper from Mazzeo et al Glu89Gln resulted quite common in Eastern Sicily, Phe64Leu in Northern, while Thr49Ala was only reported in the south, more precisely from Agrigento. Of interest, a more recent study on Western Sicily confirmed the presence of Phe64Leu in 65% of positive screening case and reported also His90Asn, Val122Ile, Ser77Phe, Val20Ala variants. Conversely, Thr49Ala and Glu89Gln were not reported. Considering a population of 5 million people in Sicily, Phe64Leu seems to be the most common variant with 28 cases reported by Mazzeo in 2015 and further 34 cases by Di Stefano in 2023. Hence, we could hypothesize a prevalence of 12.4 per million people in Sicily for Phe64Leu. However, these are only assumptions and hypothesis (for example it should be taken into account that living patients in 2015 probably are not still alive in 2023) because the exact prevalence of each variant is unknown. However, a recent multicentric Italian study on 181 patients showed that Phe64Leu is the most frequent TTR variant (54 out of 181 patients, 29.8%) in patients treated with patisiran in Italy.

The manuscript was modified as follows:

“In 2015 Mazzeo et al estimated a prevalence of hATTR in Sicily of 8.8/1000000, reporting the absence of the common Val30Met mutation, and the presence of three TTR variants (Glu89Gln, Phe64Leu, Thr49Ala) (5). In that paper Glu89Gln variant resulted quite common in Eastern Sicily, Phe64Leu in Northern, while Thr49Ala was only reported in the south, more precisely from Agrigento. However, a recent study has identified additional TTR mutations in Western Sicily, including His90Asn, Val122Ile, Ser77Phe, Val20Ala (5). Of interest, the presence of Phe64Leu was confirmed in 65% of positive screening cases and His90Asn, Val122Ile, Ser77Phe, Val20Ala variants were also reported. Conversely, Thr49Ala and Glu89Gln were not reported. Hence, Phe64Leu seems to be the most common variant circulating in Sicily with 28 cases reported by Mazzeo in 2015 and further 34 cases by Di Stefano in 2023.”

It would be helpful to know why the age >65 was chosen.

A: Thank you for this insightful comment. As a major red flags age > 65 years was chosen as in the previous study mean age in screening-positive patients was >65 (5). Also, screening negative patients were significantly younger (p=0.041*) (5). Moreover, the onset of the disease is typically after 65 years in non-endemic countries, especially for Phe64Leu phenotype. This was specified in the methods.

Spontaneous tenon ruptures are quite rare in TTRv, and more common in wtTTR, why was this chosen?

A: Thank you for this comment. In our experience, this symptom occurs also in mutated patients, especially in Val122Ile variant.

How was cardiac amyloidosis made in the ‘major red flag’ section- PYP? Cardiac MRI? Endomyocardial biopsy?

A: Thank you for this comment. Cardiac amyloidosis was diagnosed according to standard criteria and guidelines, in the case of suggestive features on scintigraphy, cardiac MRI, echo, or biopsy (Yamamoto H, Yokochi T. Transthyretin cardiac amyloidosis: an update on diagnosis and treatment. ESC Heart Fail. 2019 Dec;6(6):1128-1139. doi: 10.1002/ehf2.12518. Epub 2019 Sep 25. PMID: 31553132; PMCID: PMC6989279.). It was just a “history red flag” not a specific assessment. However, the history was collected by neurologist not a cardiologist, hence this might have caused a bias, already considered in this kind of studies. We specified this in the methods.

Did patients that met inclusion criteria have other labs drawn including prealbumin and neurofilament light chain or were they sent directly for genetic testing? Any comment about this in the algorithm to working these patients up, particularly when genetic testing may be as readily available?

A: Thank you for this comment. As described in the method section, patients meeting criteria were directly screened even in absence of other examinations. Indeed, the idea is that genetic testing is very simple with a swab test, and this may allow a very rapid diagnosis.

Hoping in positive feedback we look forward to hearing from you soon.

Kind regards,

Vincenzo Di Stefano

Reviewer 2 Report

Comments and Suggestions for Authors

The authors prospectively investigated the prevalence of hereditary transthyretin (ATTRv) amyloidosis in the municipalities of Cammarata and San Giovanni Gemini, Italy, by using a genetic screening model based on predefined “red flags” followed by “cascading screening” on first-degree relatives of patients who tested positive.

This is an interesting study providing important insights into the early diagnosis of ATTRv amyloidosis. Taking up topics of this disease is timely because new therapeutic agents for this disease, such as gene silencing and editing agents, now appear one after another.

Although I do not have any critical comments, suggestions to strengthen this manuscript are raised as follows:

  1. “ATTRv amyloidosis”, rather than “hATTR amyloidosis” is currently used based on the nomenclature recommendation from the International Society of Amyloidosis (Amyloid 2024; 31: 249-256). I would recommend using “ATTRv amyloidosis” in this manuscript.
  2. ATTRv amyloidosis was initially considered as a disease confined to endemic foci in Portugal, Japan, and Sweden. However, the presence of late-onset cases in non-endemic areas has now been recognized (Arch Neurol 2002; 59: 1771-6), indicating that ATTRv amyloidosis is more common than previously thought. This issue should be incorporated in the introduction section, by citing this study.

Author Response

Dear Editors and Reviewers,

thanks for your valuable and constructive comments. We would like you will evaluate our revised version of the manuscript for possible publication in your journal.

Reviewer #2: The authors prospectively investigated the prevalence of hereditary transthyretin (ATTRv) amyloidosis in the municipalities of Cammarata and San Giovanni Gemini, Italy, by using a genetic screening model based on predefined “red flags” followed by “cascading screening” on first-degree relatives of patients who tested positive.

This is an interesting study providing important insights into the early diagnosis of ATTRv amyloidosis. Taking up topics of this disease is timely because new therapeutic agents for this disease, such as gene silencing and editing agents, now appear one after another. Although I do not have any critical comments, suggestions to strengthen this manuscript are raised as follows:

A: Thank you for your constructive and valuable comments. We believe that screening procedures are very important to detect significant genetic variants as well as to ensure early treatments. Thank you for your great help to improve our manuscript.

“ATTRv amyloidosis”, rather than “hATTR amyloidosis” is currently used based on the nomenclature recommendation from the International Society of Amyloidosis (Amyloid 2024; 31: 249-256). I would recommend using “ATTRv amyloidosis” in this manuscript.

A: Thank you for your suggestion. We modified the manuscript replacing hATTR with ATTRv.

ATTRv amyloidosis was initially considered as a disease confined to endemic foci in Portugal, Japan, and Sweden. However, the presence of late-onset cases in non-endemic areas has now been recognized (Arch Neurol 2002; 59: 1771-6), indicating that ATTRv amyloidosis is more common than previously thought. This issue should be incorporated in the introduction section, by citing this study.

A: Thank you for your suggestion. We added this precious consideration in the introduction, citing the relevant literature suggested.

Hoping in positive feedback we look forward to hearing from you soon.

Kind regards,

Vincenzo Di Stefano

Reviewer 3 Report

Comments and Suggestions for Authors

The article by Vincenzo Di Stefano and colleagues provides details of the Genetic screening for Hereditary Transthyretin Amyloidosis in the population of Cammarata and San Giovanni Gemini through red flags and registry archives. This research article particularly suggests a genetic screening model based on predefined “red flags” followed by “cascading screening” on first-degree relatives of patients who tested positive. Overall, this is a well-written article, and the quality of the article further needs to be improved, and the following changes should be made.

The introduction section should be improved with more comprehensive background literature focusing more on the study hypothesis and outcome obtained. 

In Material and Methods section the authors have mentioned Polymerase Chain Reaction (PCR), to analyze exons 2, 3 and 4 of the TTR gene (NM_000371.4). The authors should provide further details about the primer used, its concentration, PCR condition, etc. This information must be provided for further experimental replications. 

The authors should provide a separate Conclusion section in this study.

Author Response

Dear Editors and Reviewers,

thanks for your valuable and constructive comments. We would like you will evaluate our revised version of the manuscript for possible publication in your journal.

Reviewer #3: The article by Vincenzo Di Stefano and colleagues provides details of the Genetic screening for Hereditary Transthyretin Amyloidosis in the population of Cammarata and San Giovanni Gemini through red flags and registry archives. This research article particularly suggests a genetic screening model based on predefined “red flags” followed by “cascading screening” on first-degree relatives of patients who tested positive. Overall, this is a well-written article, and the quality of the article further needs to be improved, and the following changes should be made.

A: Thank you for your constructive and valuable comments. We believe that screening procedures are very important to detect significant genetic variants as well as to ensure early treatments. Thank you for your great help to improve our manuscript.

The introduction section should be improved with more comprehensive background literature focusing more on the study hypothesis and outcome obtained.

Thank you for your suggestion. The introduction was implemented. We added this precious consideration in the introduction, citing the relevant literature suggested. Moreover, we hypothesize that these screening programs are crucial not only for detecting patients who remain misdiagnosed, but also for identifying new variants that have not yet been characterized.

In Material and Methods section the authors have mentioned Polymerase Chain Reaction (PCR), to analyze exons 2, 3 and 4 of the TTR gene (NM_000371.4). The authors should provide further details about the primer used, its concentration, PCR condition, etc. This information must be provided for further experimental replications.

Thank you for your suggestion. We have revised the Materials and Methods section to ensure that all necessary details for reproducibility are provided. Regarding the primer sequences, we have opted not to include them directly in the manuscript; however, they are available upon request. The updated text in the manuscript now states:

" DNA was extracted from a swab sample collected from each individual according to the above-mentioned criteria. To identify mutations in the TTR gene (RefSeq NM_000371.4), PCR primers (available upon request) were designed by the software Vector NTI Advance 10.3.0 (Informax Frederick, Maryland USA) to amplify exons 2, 3 and 4 with their flanking intronic regions. PCR reactions were carried out in 50 μL re-action volumes containing 200 ng genomic DNA, 1X PCR reaction buffer, 0.2 mM of each dNTP, 1 μM of each primer and 1.25 units of Taq DNA polymerase (Roche, Mannheim, Germany). The following PCR cycling conditions were used for all exons: an initial denaturation step at 94 ◦C for 6 min, followed by 35 cycles of 30 s at 94 ◦C, 30 s at 55 ◦C, 1 min at 72 ◦C, and a final extension step at 72 ◦C for 10 min.

PCR products were sequenced using the BigDye Terminator v1.1 Cycle Sequenc-ing Kit (Thermo Fisher Scientific, Vilnius, Lithuania) on a SeqStudio Flex Genetic Ana-lyzer (Applied Biosystems, Foster City, CA, USA). Patient sequence data were aligned for comparison with corresponding wild-type sequence."

The authors should provide a separate Conclusion section in this study.

Thank you for your suggestion. We added a paragraph for conclusions:

“ATTRv is a progressive and often fatal disease which should be promptly diagnosed and treated to stop progression and reduce mortality. A systematic screening for ATTRv yielded an increased recognition of the disease in our Neurological clinic. A focused approach for the screening of ATTRv-PN could lead to an earlier diagnosis and identification of asymptomatic carriers, enabling timely intervention through close clinical monitoring and early treatment initiation at symptom onset.”

Hoping in positive feedback we look forward to hearing from you soon.

Kind regards,

Vincenzo Di Stefano